# Changes in Personal Exposure to Fine Particulate Matter (PM$_{2.5}$) during the Spring 2020 COVID-19 Lockdown in the UK: Results of a Simulation Model

Ruaraidh Dobson [1],*, Douglas Eadie [1], Rachel O'Donnell [1], Martine Stead [1], John W. Cherrie [2,3] and Sean Semple [1]

1 Institute for Social Marketing and Health, Faculty of Health Sciences and Sport, University of Stirling, Stirling FK9 4LA, UK; douglas.eadie@stir.ac.uk (D.E.); r.c.odonnell@stir.ac.uk (R.O.); martine.stead@stir.ac.uk (M.S.); sean.semple@stir.ac.uk (S.S.)
2 Institute of Occupational Medicine, Edinburgh EH14 4AP, UK; j.cherrie@hw.ac.uk
3 Institute of Biological Chemistry, Biophysics and Bioengineering, Heriot Watt University, Edinburgh EH14 4AS, UK
* Correspondence: r.p.dobson@stir.ac.uk; Tel.: +44-7803-406-343

**Abstract:** Objectives: Policy responses to the COVID-19 pandemic in 2020 led to behaviour changes in the UK's population, including a sudden shift towards working from home. These changes may have affected overall exposure to fine particulate matter (PM$_{2.5}$), an air pollutant and source of health harm. We report the results of a simulation model of a representative sample of the UK's population, including workers and non-workers, to estimate PM$_{2.5}$ exposure before and during the pandemic. Methods: PM$_{2.5}$ exposure was simulated in April and August 2017–2020 for 10,000 individuals across the UK drawn from the 2011 nationwide census. These data were combined with data from the UK's ambient PM$_{2.5}$ monitoring network, time use data and data on relevant personal behaviour before and during the first stage of the pandemic (such as changes in smoking and cooking). Results: The simulated exposures were significantly different between each year. Changes in ambient PM$_{2.5}$ resulted in regional and temporal variation. People living in homes where someone smoked experienced higher exposure than those in smoke-free homes, with an increase of 4 μg/m$^3$ in PM$_{2.5}$ exposure in 2020. Conclusions: Changes in PM$_{2.5}$ exposure were minimal for most individuals despite the simulated increases in cooking activity. Those living in smoking homes (estimated to be around 11% of the UK population) experienced increased exposure to PM$_{2.5}$ during COVID lockdown measures and this is likely to have increased mortality and morbidity among this group. Government policy should address the risk of increased exposure to second-hand smoke in the event of future COVID-19-related restrictions.

**Keywords:** public health; air pollution; environmental health

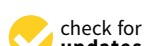

## 1. Introduction

The COVID-19 pandemic led to major changes in public policy across the world. In the UK and many other countries, non-pharmaceutical interventions to control the COVID-19 pandemic included 'lockdown' periods when travel outside of the home for non-essential reasons was forbidden by law. Consequently, the UK population spent more time at home during these periods than they would otherwise have done. The impact of this population level 'behaviour change' has been explored in terms of changes to ambient outdoor air quality, where changes varied by pollutant but tended to reflect lower vehicle traffic densities [1,2]. Personal exposure to air pollutants is much more complex [3] and depends on the amount of time the individual spends in various micro-environments (e.g., outdoors, at work, at home, within a car or public transport), the potential presence of emission sources within these environments (e.g., vehicle exhausts, cigarette smoke or

cooking fumes) and the relative toxicity of PM from those different sources [4,5]. The effect of lockdown population-level behaviour changes on personal exposure to air pollutants has not been examined. This study considers changes in personal exposure to one type of air pollution: fine particulate matter with a diameter of less than 2.5 μm ($PM_{2.5}$), which is linked to cardiovascular and pulmonary illness, including heart disease, chronic obstructive pulmonary disease [6] and stroke [7].

The UK has relatively low ambient concentrations of $PM_{2.5}$ outdoors in comparison with many other countries, particularly low- and middle-income countries [8]. The UK's annual average outdoor $PM_{2.5}$ concentration is typically 10 μg/m$^3$ [9], primarily made up of carbonaceous and secondary aerosol species [10]. Some outdoor air pollution infiltrates indoor environments, but indoor $PM_{2.5}$ concentrations are heavily affected by indoor sources. Indoor air pollution may be a greater health concern due to the fact that people spend approximately 90% of their time indoors [11] and concentrations of pollutants can be much higher because of small room volumes and poor ventilation. For example, $PM_{2.5}$ concentrations in a home living room can exceed 1000 μg/m$^3$ while someone smokes [12].

On 23 March 2020, the UK government announced substantial new public health regulations to control the spread of COVID-19. While the exact content of these regulations differed by UK nation, their main aim was consistent: to require the public to remain at home as much as possible, with only limited exceptions for essential shopping, exercise, medical need and necessary work travel. Most businesses closed during this time, and all workers who were able to work from home were encouraged to do so. This message continued throughout the rest of 2020, leading to many workers spending time in their own homes which they would otherwise have spent at work.

Due to this increased time at home, the UK population's exposure to $PM_{2.5}$ may have changed, as altered behaviour may have affected $PM_{2.5}$ concentrations both indoors and outdoors. These changes may be positive or negative: ambient $PM_{2.5}$ may have fallen due to lighter traffic but increased due to the greater use of domestic biomass burning, as has been identified in Milan [13]. This paper presents a simulation modelling approach to assess changes in personal exposure to $PM_{2.5}$ across the UK population during the first phase of the COVID-19 pandemic compared to the three previous years in order to provide data useful to assessing risks to human health.

## 2. Methods

### 2.1. Modelling Personal Exposure to $PM_{2.5}$

A model to estimate individual exposure to $PM_{2.5}$ for a cross-sectional sample of the UK population was developed using a number of data sources and the software package R v4.0.3 [14]. This model was designed to estimate an average value for the 24 h personal exposure to $PM_{2.5}$ for each of 10,000 individuals in the UK population during the months of April and August 2017, 2018, 2019 and 2020 (the study months). April was selected as it was the first full month under lockdown restrictions, while August was chosen as a comparator with generally lower outdoor ambient $PM_{2.5}$ concentrations.

### 2.2. Demographic and Time Use Data

The model was designed to produce exposure estimates for individuals across the UK based on a random selection from the UK 2011 census microdata [15], a wholly anonymised source of UK demographic information. Data from the census incorporated into the model included age, occupation, ethnicity and location (by UK Nomenclature of Territorial Units for Statistics level 1 [16]).

Data on time spent in the home pre-lockdown were extracted from the UK Time Use Survey 2014–2015 [17]. This extensive survey incorporated diary data on time use from 10,208 individuals across the UK. Arithmetic means and standard deviations of the number of minutes spent at home and number of cooking episodes per day were calculated for each work classification (whether a modelled individual worked and, if not, whether they were unemployed, a child, retired or a student) and by geographical region.

Each individual was then randomly assigned estimated time at home and number of cooking episode values, using values derived from the Time Use Survey. These were estimated by drawing from the normal distribution around mean values (truncated at 0) to provide an estimate of time spent at home and the number of episodes of cooking in a day.

### 2.3. Ambient PM$_{2.5}$ by Region

The openair R package [18] was used to download hourly outdoor PM$_{2.5}$ concentration data from all UK Automatic Urban Rural Network (AURN) [19] air pollution monitors during the study periods (pre- and post-lockdown). The AURN includes monitors using several principles of operation, including tapered element oscillation microbalances, beta attenuation monitors, filter dynamics measurement systems and others [20]. Data from the AURN were treated in line with UK Defra-provided guidance, with only verified data used in the analysis. Arithmetic means were calculated for PM$_{2.5}$ concentrations from each monitor over each study month. The geometric mean PM$_{2.5}$ was calculated for each region for each day in each study month. Data on monthly average ambient PM$_{2.5}$ concentrations by region are available in Supplementary Table S1.

### 2.4. Indoor Air

Based on previous research on air pollution in UK homes, an average infiltration factor of 0.6 was estimated for PM$_{2.5}$ from ambient air [21]; in the absence of any other source, the indoor home 24 h average concentration would have a value of 60% of the corresponding outdoor concentration. Individual infiltration factors by home were estimated by drawing from the truncated normal distribution around this mean (standard deviation = 0.1, lower bound = 0, upper bound undefined). Baseline PM$_{2.5}$ concentrations for each home were calculated for each day (00:00–23:59) as a multiple of this individual infiltration factor and the regional geometric mean ambient PM$_{2.5}$ for that day.

In addition to ambient pollution infiltration, the primary sources of PM$_{2.5}$ within homes are cooking and smoking [22,23]. For each home, cooking episodes and smoking episodes were modelled for each study month. Data on PM$_{2.5}$ concentrations in smoking homes were drawn from previous research measuring second-hand smoke (SHS) in Scottish homes [12], while data on cooking-related concentrations were calculated from data obtained in a study on non-smoking homes in Scotland [24]. These data were collected using the Dylos DC1700, a low-cost optical particle counter which has previously been validated for use monitoring SHS and which can detect particles of aerodynamic diameter 0.3 μm or greater [25,26]. The daily numbers of cooking episodes and smoking episodes were estimated, drawing from the 0-truncated normal distribution around the mean calculated from the Time Use Survey data (in the case of cooking) and a mean of 11 cigarettes smoked per day (for smoking households) was identified as the mean number of cigarettes smoked in a day in the Scottish Health Survey 2019 [27].

Indoor PM$_{2.5}$ concentrations were calculated for every ten-minute period within each participant's home. Smoking and cooking episodes were distributed randomly throughout each 24 h period. Peak PM$_{2.5}$ concentration was estimated from peak concentrations detected during cooking and smoking episodes in two pieces of previous observational research [12,24], and associated PM$_{2.5}$ concentrations over time were calculated until the concentration reached 2 μg/m$^3$ or lower. The deposition rate for both SHS and cooking fume particles (which are predominantly under 0.2 μm in diameter) was estimated at 2.0 per hour [28] and household air changes per hour were estimated at 0.65. Smoking and cooking peak data are given in Supplementary Table S2.

Based on evidence from the Scottish Health Survey 2019 [29] we estimated that smoking took place in 11% of households.

### 2.5. Changes in Behaviour during Lockdown

In order to generate estimates of how time spent at home and behaviour within the home may have changed during lockdown, we conducted a programme of qualitative

research interviewing and surveying 25 adults living in smoking households in the UK [30]. This research indicated that participants spent more time at home during the lockdown period: those in employment reported a mean increase of seven hours 49 min at home, while those not in employment spent an average of two hours 40 min more within the home environment. The number of extra minutes spent at home was estimated for each modelled individual depending on their employment status by drawing from the normal distribution around the mean additional time reported by study participants (truncated at 0 and the total number of minutes in a day, 1440).

We did not model any changes in smoking behaviour or frequency during the lockdown period, as we had limited evidence for the magnitude of these changes. Our qualitative data indicated that smoking levels showed an overall increase during lockdown [23]. To model the widely reported increase in cooking at home during lockdown [31], we increased the estimated number of cooking episodes per household by one standard deviation (from the number of episodes reported pre-lockdown in the 2014–2015 Time Use Survey).

*2.6. Exposure Modelling*

A time-weighted average $PM_{2.5}$ concentration was calculated for each individual on each day, incorporating time spent inside the home and outside. Exposure during the time spent inside the home was based on the arithmetic mean value of $PM_{2.5}$ as simulated. Exposure during the time outside of the home was assigned using the geographical region's ambient $PM_{2.5}$ concentration for that day.

*2.7. Statistical Analysis*

Summary tables of average personal $PM_{2.5}$ exposure by group population characteristics were then generated to provide detail of the changes in experiences during lockdown by each population grouping.

To determine the statistical significance of changes between the pre- and during-lockdown simulations, mean exposures in each April and August for 2017–2019 were calculated for each simulated individual and transformed logarithmically. These were compared by a pairwise t-test with similarly transformed mean exposures from April and August 2020. Statistical analysis was conducted using R v4.0.3.

*2.8. Ethical Approval*

No ethical approval was required for this study. This research used fully anonymous data from previous population surveys and observational studies. No individual was identifiable from the data used.

**3. Results**

*3.1. Overall Changes in Exposure*

Modelled exposures were significantly different when compared pairwise between each month and year ($p < 0.001$). Changes in modelled exposure between 2017–2019 and 2020 are presented in Figure 1.

Median modelled exposure declined from 11.1 µg/m$^3$ to 10.8 µg/m$^3$ between April 2017–2019 and April 2020 (median difference: −0.6 (IQR: −1.28–0.406)), but increased from 6.1 µg/m$^3$ in August 2017–2019 to 7.3 µg/m$^3$ in August 2020 (median difference: +1.23 (IQR: 0.572–1.94)).

*3.2. Changes in Exposure by Demographic Characteristics*

Significant differences in $PM_{2.5}$ in outdoor air, rather than changes in exposure to $PM_{2.5}$ indoors, drove changes in exposure between regions and over time. In particular, large increases in $PM_{2.5}$ in April 2019 (caused by a Saharan dust event [32]) caused average modelled exposures during that time to be higher than in other years (Figure 2).

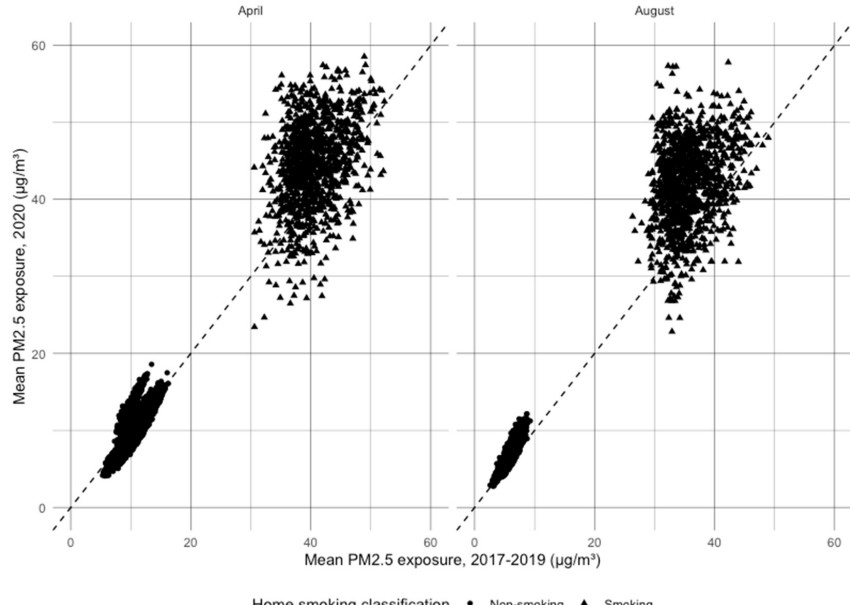

**Figure 1.** Mean modelled exposure during the months of April and August 2017–2019 vs. the same months for 2020 by participant. The group of points to the upper right (*n* = 1120) represent modelled individuals living in smoking homes, while those in the lower left are for those living in smoke-free homes (*n* = 8880). Dashed line represents identity (y = x, i.e., the personal exposure during lockdown was the same as during 2017–2019). Points to the left of the dashed line indicate an increase during lockdown.

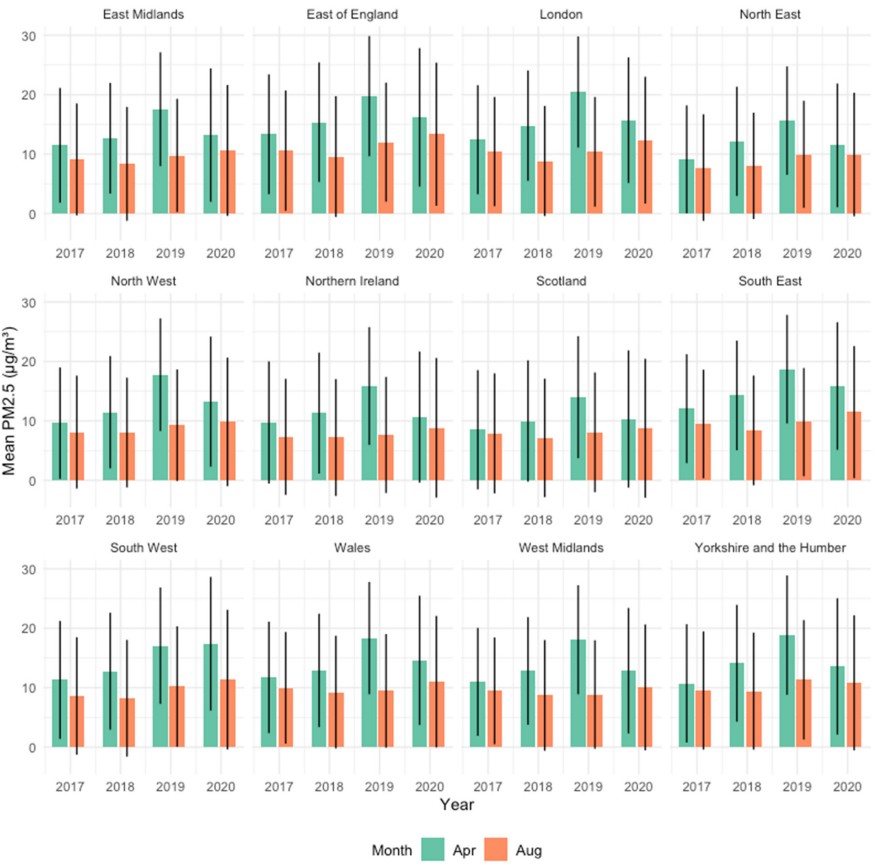

**Figure 2.** Mean PM$_{2.5}$ exposure by region in each month. Error bars represent standard deviation.

There were no significant differences in modelled exposure by other demographic characteristics, including ethnicity, gender and NS-SEC occupation code.

### 3.3. Level of Exposure by Whether Smoking Occurs in the Home

Those living in smoking homes had consistently much higher modelled exposure to $PM_{2.5}$ than those simulated as living in smoke-free homes in all modelled years (Figure 3). People living in smoking homes experienced increases of around 4 $\mu g/m^3$ in modelled mean $PM_{2.5}$ exposure during the COVID-19 lockdown periods compared to those modelled as living in non-smoking homes.

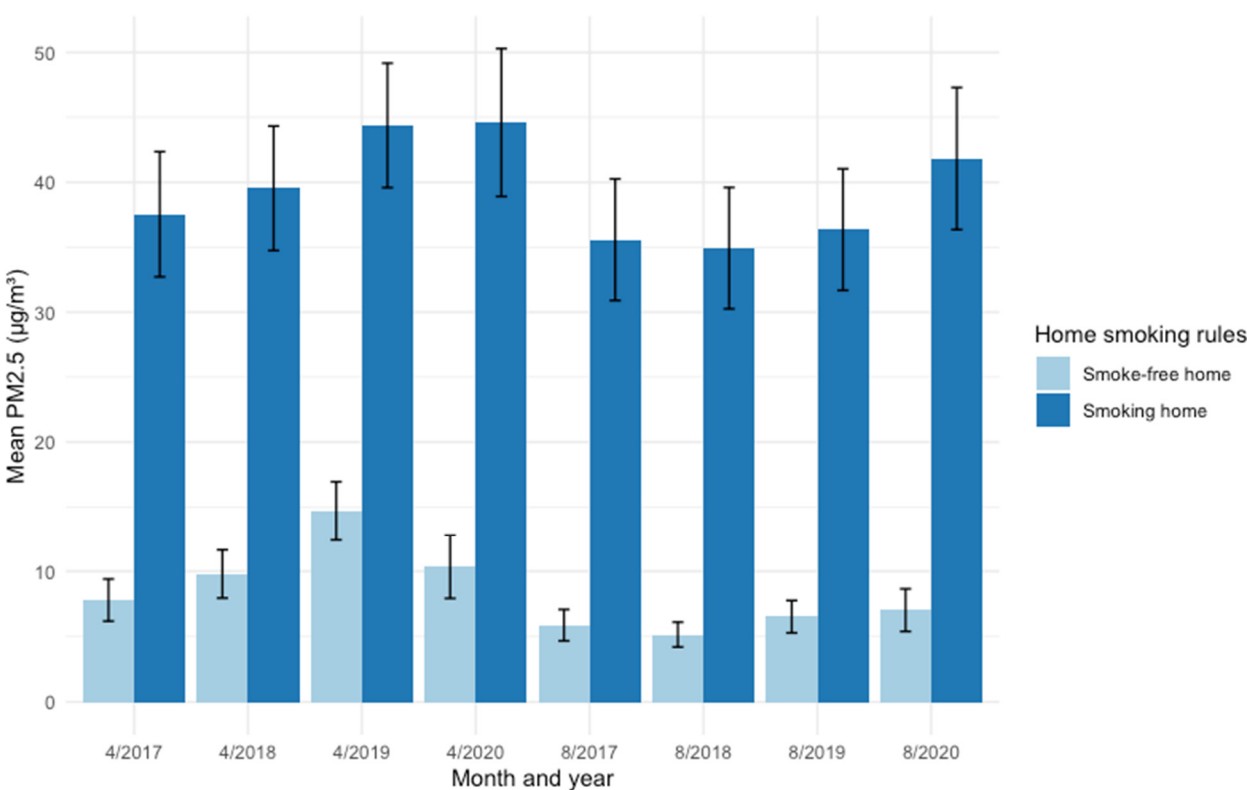

**Figure 3.** Mean $PM_{2.5}$ exposure by home smoking status in each month. Error bars represent standard deviation.

### 4. Discussion

Increases in time spent within the home setting, coupled with changes in smoking patterns and resulting exposure to second-hand tobacco smoke, are likely to have generated increases in personal exposure to $PM_{2.5}$ for those living in smoking homes during COVID-19 lockdown restrictions. However, changes in cooking activity made relatively little difference to simulated personal exposure to $PM_{2.5}$ during COVID-19 lockdowns in the UK.

The increase in personal exposure to $PM_{2.5}$ identified for those living in a smoking home was substantial and averaged about 4.5 $\mu g/m^3$ during April 2020 compared to April in 2017–2019. This represents an increase in exposure of approximately 15%, and given the scale of this change in both absolute and relative terms, it is likely to have produced changes in morbidity and mortality in line with previous exposure–response relationships for changes in $PM_{2.5}$ [33]. A simplistic assumption at the UK population level would suggest that a 4.5 $\mu g/m^3$ increase in exposure could produce about a 4% increase in all-cause mortality for the 11% of adults who live with a smoker; when applied to a single month of April 2020, this equates to approximately 100–200 additional deaths in the UK from this increased exposure to $PM_{2.5}$.

While several studies have been conducted on the effects of COVID-19 lockdowns on ambient air pollution [34–37], few have directly measured personal exposure of individuals to PM during these times. One study in Kathmandu, Nepal, reported reductions in personal exposure to $PM_{2.5}$ among US embassy staff of 50–77% [38]. However, this was likely driven by reduced exposure to Kathmandu's heavily polluted outdoor air, and comparatively wealthy diplomats were unlikely to be exposed to biomass fuel smoke (a common source of PM exposure in Nepal). In contrast, a study in rural Hunan province, China, found significantly higher $PM_{2.5}$ concentrations during the COVID-19 lockdown period than during the days immediately before it [39]. This demonstrates the importance of home circumstances in personal exposure to air pollution during lockdowns.

### 4.1. Limitations

This model was limited to smoking and cooking terms due to a paucity of data on the use of other sources of $PM_{2.5}$ in the home (such as wood-burning stoves, candles and incense). In smoking homes, cigarette smoking is likely to be the largest source of smoking-related $PM_{2.5}$ [23]. Additionally, the contribution of active smoking to personal $PM_{2.5}$ exposure was not included as a source at the level of the individual smoker. The simulation did not model changes in smoking behaviour during lockdown; given our study identified that the population spent more time at home (due to increased working from home), it is reasonable to assume that a greater proportion of cigarettes would have been smoked within the home as other typical opportunities to smoke (e.g., during travel or at/outside the workplace) would have been reduced [30]. As a result, it is likely that the simulation results are underestimates of the increases that would have been experienced for those living in a smoking home [26].

While our exposure–mortality assessment is a simplistic calculation that does not take account of many factors, including those affecting smoking behaviour [23], it does suggest that lockdown measures had the potential to produce measurable changes in respiratory and cardiovascular mortality. It is likely that morbidity (e.g., hospital admissions from exacerbations of asthma or COPD) will also have increased by similar proportions and that these may have played a role in increasing the burden on the already stretched respiratory health services. Our calculation also omits differences in the relative health effects of $PM_{2.5}$ from different sources, though in studies of this kind it may be prudent to assume that all particles have similar toxicities [40].

Due to a lack of data, we were unable to ensure that the number of cooking and smoking episodes were independent of one another. There is some evidence to suggest that UK smokers are less likely to eat home-cooked meals than non-smokers [41], which may mean that people in smoking households are less likely to be exposed to cooking-related $PM_{2.5}$. However, the size of this effect, if it exists, is unknown and it was not possible to apply this to those living in smoking homes, rather than just smokers, within the confines of this model.

As the census data were localised only to UK government regions, we were unable to estimate the effects of lockdown on air pollution exposure by the index of multiple deprivation. Due to the well-known stratification of SHS exposure by socio-economic status [42–45], we expect that individuals living in more deprived communities would in fact be more exposed to SHS.

### 4.2. Future Research

Future research in this area would ideally include more data on the impact of other indoor sources of $PM_{2.5}$, such as wood-burning stoves (now rising in popularity, with over 1 million UK households using these stoves in 2016 [46]). As the transition to home working is expected to continue as the COVID-19 pandemic recedes, it will be important to understand the ongoing changes in the home environment and exposure to indoor air pollution, including for households situated in more deprived communities. Gathering samples of $PM_{2.5}$ from these households and conducting a toxicological analysis would

greatly improve our understanding of the relative harm posed by indoor exposure under these circumstances.

## 5. Conclusions

While the COVID-19 lockdown periods resulted in substantial increases in time spent at home, this did not result in substantial changes, positive or negative, in $PM_{2.5}$ exposure in most homes in the UK. The exception was in homes where smoking took place and where personal exposure during lockdown was estimated to have increased by approximately 15% in relative terms or 4.5 $\mu g/m^3$ in absolute terms. This increase in exposure is substantial and likely contributed to measurable amounts of mortality and morbidity among the estimated 11% of UK homes where it occurred. Governments should provide targeted advice on the health harms from indoor smoking alongside any future lockdown measures or encouragement to work from home.

**Supplementary Materials:** The following supporting information can be downloaded at: https://www.mdpi.com/article/10.3390/atmos13020273/s1, Table S1: Mean $PM_{2.5}$ concentration by month and UK nation/English region. Geometric means were calculated for all regions and overall by month and year, and then the arithmetic mean calculated for each month in the analysis period in 2017–2019. Table S2: $PM_{2.5}$ peak height distributions for smoking and cooking episodes.

**Author Contributions:** Conceptualization, R.D., J.W.C. and S.S.; methodology, R.D., J.W.C. and S.S.; software, R.D.; validation, J.W.C. and S.S.; data curation, R.D., R.O., D.E. and S.S.; writing—original draft preparation, R.D.; writing—review and editing, D.E., R.O., M.S., J.W.C., S.S.; visualization, R.D.; supervision, S.S.; project administration, S.S. and M.S.; funding acquisition, M.S. All authors have read and agreed to the published version of the manuscript.

**Funding:** This research was funded by the National Institute for Health Research PH-PRU as Project 08 (PHSEZQ47-22-C).

**Institutional Review Board Statement:** Not applicable.

**Informed Consent Statement:** Not applicable.

**Data Availability Statement:** The data presented in this study include a mix of publicly available datasets (e.g., UK Time Use Survey data), datasets available from the corresponding author (e.g., data on $PM_{2.5}$ associated with smoking and cooking) and 3rd party data (e.g., UK Census microdata) to which restrictions apply for privacy reasons. Please contact the corresponding author for further details.

**Conflicts of Interest:** The authors declare no conflict of interest.

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
