# Peer review of "Changes in Personal Exposure to Fine Particulate Matter (PM2.5) during the Spring 2020 COVID-19 Lockdown in the UK: Results of a Simulation Model"

_atmosphere, doi:10.3390/atmos13020273_

Round 1

Reviewer 1 Report

Reviewer’s comments:

In this research the policy response to the Covid-19 pandemic was described with the environment study of fine particulate matter (PM2.5) typically reached as PM2.5 average concentration of 10μg/m3 outdoor and usually reached as PM2.5 average concentration of exceed 1,000μg/m3 in a home living room with second-hand smoke and cooking activity around 24 .11% of the UK population.

In this paper, the simulation modeling approach to assess the changes in personal exposure to PM2.5 during the first phase of the COVID-19 pandemic has been completed and compared to the three previous years using a number of data sources and the software package R v4.0.3.

  1. The ANOVA test of different outdoor geography environments such as the environment temperature or other particle population should be discussed in "4.Discussion" section.
  2. The tests of independence between dependent variable parameters should be explored and described in detail in "4.Discussion" section.
  3. The manuscript typing should be confirmed such as the duplicating typing of “using” on line 86, and the “Exposure by home smoking rules” on line 201.

Reviewer 2 Report

The paper tries to investigate the impacts of COVID-19 lockdown on indoor air pollution (with a focus on PM2.5 levels) in UK. The authors used a model combining different parameters (such as ambient PM2.5 levels, demographic data, etc.) to estimate the indoor PM2.5 exposure. The work is interesting (given that the recent huge race in COVID-19 associated aerosol research was mostly focused on outdoor air pollution), and most of the interpretations are sound and in my opinion it merits publication in the Atmosphere journal. However, further details are needed in some sections of the manuscript before reaching the format and content standard for the readership of Atmosphere. Accordingly, I recommend minor revisions following the comments below.

Comments:

1) Lines 54-56: It is worth adding that the toxicity of PM2.5 also varies depending on the emission source, which adds to the complexity of modeling the exposure.

(doi.org/10.1038/s41586-020-2902-8; doi.org/10.1016/j.scitotenv.2016.06.231)

2) Lines 61-69: This paragraph needs further details about the PM2.5 sources in the UK.

(doi.org/10.1016/j.atmosenv.2004.08.037; doi.org/10.4209/aaqr.2015.09.0537)

3) Lines 78-79: Even in some cases, the ambient PM2.5 levels did not undergo a significant decrease as stay-home policies have led to augmented levels of emissions from residential sector. For example, increased emissions from residential heating (biomass burning) have been observed in Italy and China (as the epicenters of COVID-19 in Europe and Asia). This should be added to the paragraph, which further necessitates the investigation of indoor air pollutants.

(doi.org/10.1016/j.scitotenv.2020.139542; doi.org/10.1016/j.scitotenv.2020.143582)

4) Lines 110-112: Please add details about the PM2.5 measurement apparatus, its LOD, and whether any data correction techniques have been applied (such as removing outliers, or how to treat the missing data).

5) Lines 116-133: According to the text, 0.6*ambient PM2.5 was used as the baseline, and emissions from cooking/smoking has been added to the baseline levels. It is difficult to follow each step, so I would suggest adding a “summary statistic” table and providing the mean/range for each parameter (for example the smoking data based on the mentioned references), then adding them together.

Having said that, the table can demonstrate how ambient PM2.5 has been decreased as a result of lockdown conditions (and accordingly for baseline PM2.5, because it’s basically 0.6*ambient PM2.5). Then we can compare it with the increase emissions from cooking and smoking.

6) Lines 183-184: Please add more details regarding the differences between each pair of the data. For instance, the percentage change in PM2.5 during April between 2017-2019 and 2020, etc.

7) Line 261-267: The best method to determine the impact of COVID lockdown on indoor air pollution would be to collect filter samples and chemically/toxicologically analyze them. I would suggest adding it in the text for future research.

Edits:

1) Line 86: extra “using”

Reviewer 3 Report

Review of “Changes in Personal Exposure to Fine Particulate Matter (PM2.5) During the Spring 2020 Covid-19 Lockdown in the UK: Results of a Simulation Model”.

The authors discussed the changes in Personal Exposure to PM2.5 During the Spring 2020 Covid-19 Lockdown in the UK in this study. I think the topic is important and interesting. I have some concerns as follows.

1. The indoor PM2.5 levels are from the infiltration of the outdoor PM2.5, indoor cooking and smoking. I suggest that the authors explain clearly how to quantify the contribution of cooking and smoking. Is there any real value observation in this process?

2. Working from home will increase cooking and smoking, does this have any effect on the conclusion? 

3. Line 22: “4µg/m3” needs a blank, and also modify it for the whole paper.

4. Figure 2: Please give more explanations for the new finding in the main text.

5. Line 157: remove the extra “.”.

6. Line 201: “Exposure by home smoking rules” means what?

Round 2

Reviewer 1 Report

The format in References section should be confirmed following the Journal style.

Reviewer 2 Report

My comments have been addressed.